# Effect of diurnal intermittent fasting during Ramadan on ghrelin, leptin, melatonin, and cortisol levels among overweight and obese subjects: A prospective observational study

Natheer Al-Rawi[1], Mohamed Madkour[2], Haitham Jahrami[3,4], Dana Salahat[2], Fatima Alhasan[2], Ahmed BaHammam[5,6], Mo'ez Al-Islam Faris[7] *

1 Department of Oral and Craniofacial Health Sciences, College of Dental Medicine/Research Institute of Medical and Health Sciences (RIMHS), University of Sharjah, Sharjah, UAE, 2 Department of Medical Laboratory Sciences, College of Health Sciences/Research Institute of Medical and Health Sciences (RIMHS), University of Sharjah, Sharjah, UAE, 3 Rehabilitation Services, Periphery Hospitals, Ministry of Health, Manama, Bahrain, 4 College of Medicine and Medical Sciences, Arabian Gulf University, Manama, Bahrain, 5 Department of Medicine, College of Medicine, University Sleep Disorders Center, King Saud University, Riyadh, Saudi Arabia, 6 The Strategic Technologies Program of the National Plan for Sciences and Technology and Innovation in the Kingdom of Saudi Arabia, Riyadh, Saudi Arabia, 7 Department of Clinical Nutrition and Dietetics, College of Health Sciences/Research Institute of Medical and Health Sciences (RIMHS), University of Sharjah, Sharjah, UAE

* mfaris@sharjah.ac.ae, moezfaris@hotmail.com

## Abstract

### Background

Levels of cortisol, melatonin, ghrelin, and leptin are highly correlated with circadian rhythmicity. The levels of these hormones are affected by sleep, feeding, and general behaviors, and fluctuate with light and dark cycles. During the fasting month of Ramadan, a shift to nighttime eating is expected to affect circadian rhythm hormones and, subsequently, the levels of melatonin, cortisol, ghrelin, and leptin. The present study aimed to examine the effect of diurnal intermittent fasting (DIF) during Ramadan on daytime levels of ghrelin, leptin, melatonin, and cortisol hormones in a group of overweight and obese subjects, and to determine how anthropometric, dietary, and lifestyle changes during the month of Ramadan correlate with these hormonal changes.

### Methods

Fifty-seven overweight and obese male (40) and female (17) subjects were enrolled in this study. Anthropometric measurements, dietary intake, sleep duration, and hormonal levels of serum ghrelin, leptin, melatonin, and salivary cortisol were assessed one week before the start of Ramadan fasting and after 28 days of fasting at fixed times of the day (11:00 am-1:00 pm).

### Results

At the end of Ramadan, serum levels of ghrelin, melatonin, and leptin significantly (*P*<0.001) decreased, while salivary cortisol did not change compared to the levels assessed in the pre-fasting state.

**Data Availability Statement:** Data underlying the study is available on the data repository: https://figshare.com/s/31c1c984b1e6e3f63d07.

**Funding:** This work was supported by a Vice-Chancellor Research and Graduate Studies Office/University of Sharjah grant no. (VCRG/R1061/2016).

**Competing interests:** The authors declare they have no competing interests.

**Abbreviations:** BIA, Bioelectrical impedance analysis; BMI, Body mass index; DIF, Diurnal intermittent fasting; ELISA, Enzyme-linked immunosorbent assay; HC, Hip circumference; HDL, High-density lipoprotein cholesterol; LDL, Low-density lipoprotein cholesterol; METs, Metabolic equivalents; SD, Standard deviation; STROBE, Strengthening the Reporting of Observational Studies in Epidemiology; UAE, Sharjah, United Arab Emirates; WC, Waist circumference..

## Conclusions

DIF during Ramadan significantly altered serum levels of ghrelin, melatonin, and serum leptin. Further, male sex and anthropometric variables were the most impacting factors on the tested four hormones. Further studies are needed to assess DIF's impact on the circadian rhythmicity of overweight and obese fasting people.

## Introduction

Circadian rhythms are mental, physical, and behavioral changes that follow a daily cycle, which respond primarily to light and darkness in an organism's environment [1]. Circadian rhythm has a direct effect on metabolic processes and the regulation of energy balance [2]. Certain behaviors such as the timing of food abstinence and intake may affect circadian rhythms [3–6]. Levels of cortisol, melatonin, ghrelin, and leptin are highly correlated with circadian rhythmicity; the levels of these hormones are affected by sleep, feeding, and general behaviors, and fluctuate according to the light-dark cycle [7].

Recurrent circadian fasting during Ramadan is a form of diurnal intermittent fasting (DIF) or time-restricted feeding that is practiced by more than 1.5 billion Muslims every year. This special form of religious fasting is characterized by its duration and continuity (no break between fasting days) [8–11]. Such type of fasting involves complete abstinence from food and drink, even water, from dawn to sunset. During Ramadan, several changes have been reported in dietary habits, meal frequency, and timing [12–14], and sleep patterns [15, 16]. The extent of these changes differs culturally and is associated with variable intakes of fatty and sweet foods and beverages [17, 18]. Such dietary changes are resulting in variable effects on body composition [10], metabolic syndrome components [19], genetic expressions [11], and oxidative stress and inflammatory markers [20].

During DIF of Ramadan, two to three meals are consumed after sunset: a light meal or breakfast at the moment of declaration of sunset prayer time (*Athan Al-Maghrib*), dinner following nightly prayer in some countries (about one to three hours after sunset), and a predawn meal (*Suhur*). People in other countries take one large meal at the sunset breakfast and another meal pre-dawn, with several snacks and sweets in the hours between these mealtimes. In both cases, food consumption times are shifted to nighttime hours. This sudden and drastic shift in the timing of food intake partly inverts the normal circadian patterns, leading to a disruption of the biological clocks and rhythms of fasting people [4]. Both feeding and sleep/wake cycles are under strict hypothalamic control. As such, abnormal patterns in one axis may affect the other, resulting in metabolic dysfunction [21]. Increasing caloric intake at night may affect the circadian rhythm and result in a loss of the temporal control (circadian disruption) of metabolic processes during fasting [22, 23], as circadian rhythms are impacted by the timing of meals and blood glucose levels [24–26].

During the DIF of Ramadan, drastic changes in the timing and frequency of fluid and food intake have been reported [27], which contribute to chronobiological changes in the circadian rhythms that regulate body temperature, blood glucose, cortisol, and melatonin levels, and day- and night-time wakefulness [22]. Bogdan and colleagues have previously reported that DIF during Ramadan, along with the concomitant changes in social habits, psychology, and sleep schedules induce changes in the rhythmicity of levels of several hormones such as cortisol, melatonin, and leptin [28, 29]; however, the authors did not account for the body weight of the study participants, nor did they examine how changes in anthropometric measures, diet, and lifestyle during Ramadan impact these hormonal changes.

To date, no study has assessed changes in the concentrations of these hormones in overweight/obese subjects throughout the fasting month of Ramadan. Thus, it was prudent to determine whether obese subjects differed from lean subjects in terms of changes in circadian rhythmicity during the fasting month of Ramadan. Changes in sleep and circadian rhythmicity are associated with obesity and its related biochemical, anthropometric, and dietary markers [30, 31]. It was hypothesized that the practice of DIF during Ramadan, with its accompanying dietary and lifestyle changes, would variably affect the levels of hormones that regulate feeding patterns, stress responses, and sleep among overweight and obese subjects. Therefore, the present study aimed to examine the effect of DIF during Ramadan on serum levels of ghrelin, leptin, melatonin, and salivary cortisol levels in a group of overweight and obese subjects, and to determine how anthropometric, dietary, and lifestyle changes correlate with hormonal fluctuations.

## Methods

### Subjects

The study protocol was approved by the Research Ethics Committee of the University of Sharjah (Reference no: REC-16-05-11-01) and was carried out in accordance with the tenets of the Declaration of Helsinki. All enrolled subjects provided written informed consent before starting the study. Subjects were recruited using social media outlets. The participation was completely voluntary, with no monetary or non-monetary incentives were given to the participants. All the subjects were Arab residents living in Sharjah, United Arab Emirates (UAE). A total of 64 subjects visiting the University Hospital of Sharjah in the UAE were screened. The inclusion criteria were adult male or female Muslims who were overweight or obese (BMI > 25 kg/m$^2$) who were willing to fast during the month of Ramadan. We excluded subjects with a history of endocrine or cardiovascular diseases or diabetes. We also excluded subjects who took any prescribed medication one week before Ramadan and excluding those who took any medication during the study period. Moreover, pregnant women, any subject enrolled in a weight-management program within the month before Ramadan, and those with a history of bariatric surgery were excluded.

None of the study subjects reported having had sleep problems and all participants had a regular sleep/wake schedule. Criteria of regular sleep/wake schedule include the lack of erratic sleep patterns, and having a fixed schedule of sleep/awake without continuous interruptions, thus enabling meeting the body needed sleep hours each day, and avoiding feeling with daytime sleepiness. No subjects were participating in any dietary program before the start of the study, nor were any practicing fasting as routine, voluntary rituals before the month of Ramadan.

### Study design

This prospective study was conducted one week before and during Ramadan of the lunar year (1438 *Hijri*) from May 26th to June 27th 2017. Data were collected one week before Ramadan (baseline, or T1) and at the end of the fourth week of Ramadan (after completing 28 consecutive days of fasting, or T2). During Ramadan, study subjects abstained from all food and drink (including water) from dawn to sunset. The daily fasting duration during this study was approximately 15 h. Each participant served as his/her internal control, as each subject's values before Ramadan (T1) were compared to those at the end of Ramadan (T2). No special dietary recommendations were given to the study subjects, and all participants were instructed to continue their regular diet during non-fasting hours. Notably, females are exempted from fasting during Ramadan while menstruating. Thus, the fasting period for pre-menopausal female

subjects ranged from 23 to 25 days, whereas the fasting period for the male subjects ranged from 28 to 30 days. Since physical exercise may interfere with the subjects' body compositions and biochemical measurements by the end of Ramadan, study subjects were instructed not to alter their habitual physical exercise levels before or during Ramadan.

## Anthropometric assessment

One week before the month of Ramadan (T1), subjects were assessed in the late morning (11:00 am-1:00 pm) after fasting for eight to ten hours. At the end of the month of Ramadan (T2), subjects were again evaluated in the late morning (11:00 am-1:00 pm) after fasting for eight to ten hours following the previous night's meal (*Suhur*). Bodyweight and body composition (including visceral fat assessment) were evaluated, and blood samples were drawn at the two-time points (T1 and T2). Body weight was measured in light clothing to the nearest 0.1 kg (±0.1 kg) using a balance beam scale (Detecto, Webb City, MO, USA). Fat mass, fat-free mass, and total body water were measured using direct segmental multi-frequency bioelectrical impedance analysis (DSM-BIA) (Tanita MC-980, Tokyo, Japan). Validation and high accuracy of DSM-BIA in the quantification of total body and segmental body composition had been reported elsewhere [32, 33]. As per the manufacturer's instructions, all accessories, metals, and/or jewelry were removed before conducting BIA, and each subject was instructed to excrete excess bodily fluids through urination before conducting the BIA measurements. As all subjects were fasting for eight to ten hours before each of the two-time points, the impact of hydration and physical exercise on BIA measurements was minimized, thus reducing intra-individual variability. Height was measured without shoes or head cover using a wall-mounted stadiometer (±0.1 cm). BMI was calculated based on each subject's height and weight. Waist circumference (WC) and hip circumference (HC) were measured to the nearest 1.0 cm using a non-stretchable measuring tape (Seca, Hamburg, Germany). Neck circumference was measured in the midway of the neck, between the mid-cervical spine and mid anterior neck, to within 1 mm, with non-stretchable plastic tape with the subjects standing upright. In men with a laryngeal prominence (Adam's apple), it was measured just below the prominence [34].

## Blood sampling and lipid and hormone assays

Venous blood (10 mL) samples were collected after measuring blood pressure. Venipuncture collection was done after eight to ten hours of fasting at both time points (the baseline, T1, and at the end of the fourth week of Ramadan, T2, between 11:00 am and 1:00 pm for both time-points) to minimize the effect of time on the measured variables. Coded blood samples were centrifuged at 2500 rpm for 15 minutes at 4°C within an hour of their collection, then aliquoted and stored at -80°C until analysis. Serum ghrelin (USCN, CEA99IHu), leptin (Arigo Biolaboratories, ARG80831, EO7), melatonin (Abcam, Ab213978), and salivary cortisol (Salimetrics, ER HS 1–3002) levels were measured using enzyme-linked immunosorbent assay (ELISA) kits according to the manufacturer's instructions. A fully automated clinical chemistry analyzer (Adaltis Pchem1, Rome, Italy) was used to quantify fasting plasma glucose levels and serum lipid profiles (total cholesterol, LDL-cholesterol, HDL-cholesterol, and triglycerides). Blood pressure, oxygen saturation, and heart rate were measured using a digital blood pressure monitor (GE Healthcare, Chicago, USA), with subjects in a seated position after five minutes of rest before drawing blood.

Unstimulated saliva was collected and processed according to the Salimetrics manufacture manual. For saliva collection, the fasting donors already came while avoided eating, drinking, and smoking, and were instructed not to use oral hygiene products for at least 1 hour before their hospital visit. Spit samples were collected by asking volunteers to forcefully spit saliva

(not sputum) into the sterile plastic saliva collection containers through direct spitting. The collected samples were centrifuged at 1500 gx for 15 min at 4°C within one hour of saliva collection. After centrifuge, the supernatants were immediately separated by sterile pipettes, aliquoted into smaller volumes, and stored at -80°C deep freezer until used for cortisol assay. To avoid hormone degradation thawed saliva samples were used once.

## Dietary intake assessment

Dietary intake was assessed by the 24-hour recall on three days (one weekend day and two weekdays) at the two-time points (T1 and T2) by trained nutritionists. Two-dimensional food models were used to help the study subjects determine approximate portion sizes. Dietary caloric and macronutrient intakes were estimated using The Food Processor software, version 10.6 (ESHA Research, Salem, OR, USA). Validity and reliability of 24-hour recall in assessing dietary and caloric intake are reported elsewhere [35–37].

## Physical activity assessment

General physical activity levels were assessed using the Dietary Reference Intakes: The Essential Reference for Dietary Planning and Assessment classification system developed by the Food and Nutrition Board, Institute of Medicine, USA [38]. The activity level was considered sedentary when the participant tended to spend most of the daytime hours performing basic daily life activities without being engaged in any additional physical exercise. The activity level was considered low when the participant spent the daytime hours performing daily life activities in addition to spending 30–60 minutes a day engaged in moderate-intensity physical exercise (3–6 metabolic equivalents, METs). An active participant was one whose daytime hours were spent performing daily life activities in addition to spending at least 60 minutes a day performing moderate-intensity physical exercise. A very active participant was one whose daytime hours were spent performing daily life activities in addition to at least 120 minutes a day of moderate-intensity physical exercises or 60 minutes of vigorous exercises (>6 METs) [38].

## Sleep duration

Total sleep duration (hours/day) was assessed using the Sleep Pattern questions of the Stanford Health Care Sleep Questionnaire before and at the end of the fasting month of Ramadan.

## Statistical analyses

Analyses were conducted and results were reported following Strengthening the Reporting of Observational Studies in Epidemiology (STROBE) guidelines [39]. The primary outcome measures were the changes in concentrations of the four hormones between the pre-fasting baseline and post-fasting timepoints. From a power analysis, we calculated that 51 subjects would be required to provide 80% power to detect a significant difference of 5% in hormonal changes between the baseline and post-fasting time points, using a two-tailed, paired-samples t-test with $\alpha = 0.05$. We estimated a dropout rate of 10%. Thus, we planned to enroll a total of 57 subjects. The statistical analyses were performed using Stata software (v.13.1 StataCorp., TX, USA). Results were expressed as mean ± standard deviation (SD). Two-tailed paired sample t-tests were used to compare within-subject changes from baseline. Spearman's correlation coefficients were calculated to assess the relationships between the concentrations of the four hormones and age, sex, sleep duration, and dietary intake. Differences were considered significant at $P < 0.05$.

## Results

A total of 64 subjects were initially screened. One participant was excluded due to a history of diabetes, two subjects were excluded for taking medications shortly before, and during the month of Ramadan month, one was excluded due to scheduling conflicts, and three were excluded for following weight-reducing diets. Fifty-seven subjects (17 females and 40 males) with an average age of 38.4 ± 11.2 years and a BMI of 29.9 kg/m$^2$ completed the study. The vast majority of the participants (89.5%) lived a sedentary lifestyle, with no physical activities before or during the month of Ramadan.

### Anthropometric and hemodynamic characteristics

Changes in anthropometric and hemodynamic measurements are shown in Table 1. By the end of Ramadan, body weight, BMI, neck, waist (*P<0.05*) and hip circumferences, body fat percentages and fat masses, and oxygen saturation levels decreased significantly (*P<0.0001*) from pre-fasting baselines.

### Sleep duration and dietary intakes

The total number of sleep hours (*P<0.001*) and dietary cholesterol intake significantly (*P<0.0001*) decreased during the month of Ramadan compared to pre-fasting levels, while total sugars (*P<0.0001*), polyunsaturated fats (*P<0.05*), vitamins C (*P<0.001*) and E (*P<0.05*), omega-3 fatty acids (*P<0.0001*), and lycopene (*P<0.05*) levels significantly increased (Table 2).

### Plasma levels of lipids and hormones

High-density lipoprotein levels significantly (*P<0.0001*) increased by the end of the fasting month compared to the pre-fasting levels, while low density-lipoproteins significantly

**Table 1. Anthropometric and hemodynamic measures before (T1) and at the end of Ramadan DIF (T2) for the whole population of male and female overweight and obese subjects.**

| Parameter | T1 | | T2 | | P-value |
|---|---|---|---|---|---|
| | Mean | SD | Mean | SD | |
| Weight (kg) | 88.3 | 16.2 | 86.7 | 15.7 | ** |
| BMI (kg/m$^2$) | 29.90 | 5.02 | 29.4 | 4.9 | ** |
| Neck circumference (cm) | 38.7 | 3.5 | 38.0 | 3.5 | ** |
| Waist circumference (cm) | 98.6 | 13.7 | 97.2 | 13.0 | * |
| Hip circumference (cm) | 110.1 | 9.5 | 108.6 | 8.9 | ** |
| Total body water (kg) | 44.0 | 7.3 | 43.8 | 7.0 | NS |
| Body fat percentage area (%) | 29.5 | 7.1 | 28.6 | 7.3 | ** |
| Fat mass (kg) | 26.5 | 9.5 | 25.3 | 9.4 | ** |
| Fat-free mass (kg) | 61.8 | 10.4 | 60.9 | 11.0 | NS |
| Muscle mass (kg) | 58.7 | 9.9 | 58.4 | 9.6 | NS |
| Visceral fat surface area (cm$^2$) | 100.00 | 48.6 | 96.8 | 46.1 | NS |
| Systolic blood pressure (mmHg) | 123.7 | 12.2 | 125.6 | 12.8 | NS |
| Diastolic blood pressure (mmHg) | 72.4 | 9.0 | 73.5 | 10.3 | NS |
| Resting heart rate (bpm) | 71.7 | 9.3 | 72.1 | 9.3 | NS |

*P≤0.05

** *P≤0.0001*, using paired sample *t*-test to compare the end of Ramadan diurnal intermittent fasting (DIF) to pre-fasting baseline. NS, non-significant.

**Table 2. The average number of daily sleep hours and dietary intake of selected macro- and micronutrients measured before (T1) and at the end of Ramadan DIF (T2) for the whole population of male and female overweight and obese subjects.**

| Parameter | T1 | | T2 | | P-value |
|---|---|---|---|---|---|
| | Mean | SD | Mean | SD | |
| Sleep duration (h/day) | 8.3 | 0.8 | 6.4 | 1.03 | ** |
| Total caloric intake (kcal/day) | 2,123.1 | 754.2 | 2,150.4 | 847.1 | NS |
| Calories from fats (kcal/day) | 694.5 | 396.4 | 687.1 | 366.2 | NS |
| Total carbohydrates (g/day) | 253.7 | 94.8 | 282.7 | 122.9 | NS |
| Total sugars (g/day) | 65.9 | 31.3 | 107.7 | 53.7 | *** |
| Total fats (g/day) | 77.3 | 44.1 | 76.8 | 41.1 | NS |
| Saturated fats (g/day) | 23.5 | 12.56 | 22.7 | 12.2 | NS |
| Monounsaturated fats (g/day) | 20.2 | 11.2 | 23.05 | 13.8 | NS |
| Polyunsaturated fats (g/day) | 10.8 | 8.1 | 16.3 | 16.8 | * |
| Trans fats (g/day) | 0.50 | 0.8 | 0.5 | 1.21 | NS |
| Cholesterol (mg/day) | 395.9 | 176.9 | 272.6 | 181.9 | *** |
| Vitamin C (mg/day) | 73.5 | 50.6 | 97.4 | 66.7 | ** |
| Vitamin E (mg/day) | 5.8 | 3.6 | 8.5 | 9.4 | * |
| α-carotene (μg/day) | 12.5 | 22.8 | 16.2 | 33.3 | NS |
| β-carotene (μg/day) | 393.9 | 658.0 | 576.2 | 938.6 | NS |
| Omega-3 fatty acids (mg/day) | 0.7 | 0.6 | 1.8 | 2.3 | *** |
| Omega-6 fatty acids (mg/day) | 7.8 | 7.1 | 10.1 | 10.7 | NS |
| Lycopene (μg/day) | 1,484.4 | 3,493.4 | 4,234.4 | 9,700.6 | * |
| Selenium (μg/day) | 85.8 | 48.5 | 71.0 | 47.2 | NS |
| Total water intake (mL/day) | 1397.0 | 690.3 | 1,518.6 | 808.2 | NS |

*P≤0.05

** P≤0.001

*** $P \leq 0.0001$, using paired sample $t$-test comparing the end of Ramadan DIF versus pre-fasting baseline. NS, non-significant.

($P<0.0001$) decreased. At the end of the fasting month, serum levels of ghrelin, leptin, and melatonin were significantly ($P<0.001$) decreased compared to baselines by 19%, 13%, and 39%, respectively; however, salivary cortisol levels were not significantly changed at the end of the fasting month (Table 3).

## Correlation between hormone levels, sleep duration, dietary intake, and basic and anthropometric characteristics

Correlations between hormone levels and basic demographic characteristics (age and sex), sleep duration, total caloric intake, and anthropometric variables of fasting overweight and obese subjects are shown in Table 4, before (T1) and at the end (T2) of one month of intermittent fasting during Ramadan using the Spearman correlation rank equation. Male sex was significantly associated with changes in serum ghrelin and leptin levels before ($r = -0.32$, $P<0.05$; $r = 0.42$, $P<0.001$, respectively) and after ($r = 0.38$, $P<0.001$; $r = 0.42$, $P<0.001$, respectively) Ramadan fasting. BMI, on the other hand, was significantly associated with changes in the levels of serum leptin before ($r = 0.38$, $P<0.001$) and after ($r = 0.38$, $P<0.001$) fasting. Only the total body fat percentage was significantly associated with serum leptin levels. Sleep duration was not associated with ghrelin, leptin, melatonin, or cortisol levels. Waist circumference was significantly associated with serum ghrelin levels after Ramadan ($r = -0.32$, $P<0.05$), and hip circumference was significantly associated with changes in serum leptin levels before ($r = 0.40$, $P<0.001$) and after ($r = 0.44$, $P<0.001$) Ramadan fasting month (Table 4).

**Table 3. Blood glucose, lipid profile, serum hormone (leptin, ghrelin, and melatonin), and salivary cortisol levels before (T1) and at the end of Ramadan DIF (T2) in overweight and obese subjects.**

| Parameter | T1 | | T2 | | P-value |
|---|---|---|---|---|---|
| | Mean | SD | Mean | SD | |
| FBG (mg/dL) | 99.9 | 20.5 | 105.7 | 24.3 | NS |
| TC (mg/dL) | 173.9 | 30.1 | 175.9 | 35.8 | NS |
| TG (mg/dL) | 93.6 | 53.9 | 97.4 | 44.10 | NS |
| HDL (mg/dL) | 45.6 | 6.8 | 58.4 | 11.70 | *** |
| LDL (mg/dL) | 109.6 | 32.6 | 98.0 | 34.0 | *** |
| Serum ghrelin (pg/mL) | 566.0 | 292.1 | 460.0 | 201.8 | ** |
| Serum leptin (pg/mL) | 18.4 | 12.4 | 16.0 | 12.1 | ** |
| Serum melatonin (pg/mL) | 1.2 | 260.5 | 748.3 | 381.1 | ** |
| Salivary cortisol (pg/mL) | 2.2 | 0.40 | 2.1 | 0.4 | NS |

** $P \leq 0.001$

*** $P \leq 0.0001$ using paired sample *t*-test to compare the end of Ramadan diurnal intermittent fasting (DIF) to pre-fasting baselines.

FBG, Fasting blood glucose; HDL, High-density lipoproteins; LDL, Low-density lipoproteins; NS, non-significant; TC, Total cholesterol; TG, Triglycerides

## Discussion

In this prospective observational study of overweight and obese subjects who intermittently fasted for a month during the observation of Ramadan, significant reductions were observed in anthropometric indicators of body weight, BMI, body fat percentage, and neck, waist, and hip circumferences, and there was a significant reduction in total sleep hours at the end of the fasting month compared to the subjects' pre-fasting levels. Further, serum levels of the hormones leptin, ghrelin, and melatonin were significantly reduced at the end of the fasting month compared to the pre-fasting state, though salivary cortisol levels were not significantly altered. To the best of our knowledge, the effect of DIF during Ramadan on these hormones in overweight and obese subjects has not been studied previously.

In their critical review [3], Almeneessier and colleagues extensively assessed the current literature concerning the impact of Ramadan DIF on sleep, daytime fatigue or alertness, and chronobiological markers in healthy, lean, not overweight or obese, people. However, due to the excessive adiposity, chronic loss of hormonal oscillations is associated with obesity while their body homeostasis is profoundly affected by circadian rhythms of corticosteroid secretion [40]. Therefore, the current work tried to fill the gap in knowledge with this respect.

The significant changes reported in HDL and LDL at the end of the fasting month in comparison with the pre-fasting levels are consistent with the findings of the systematic reviews

**Table 4. Correlation analysis between hormone levels and demographic characteristics (age and sex), sleep, total caloric intake, and anthropometric variables of fasting overweight and obese subjects before (T1) and at the end (T2) of DIF during Ramadan.**

| Hormone | Age | | Sex (Male) | | Body mass index (BMI) | | Waist circumference | | Hip circumference | | Body fat % | |
|---|---|---|---|---|---|---|---|---|---|---|---|---|
| | T1 | T2 | T1 | T2 | T1 | T1 | T1 | T2 | T1 | T2 | T1 | T2 |
| Serum ghrelin | r = - 0.08 | r = - 0.10 | r = - 0.32* | r = 0.38** | r = - 0.25 | r = - 0.25 | r = 0.27* | r = - 0.32* | r = - 0.16 | r = - 0.20 | r = 0.07 | r = 0.04 |
| Serum leptin | r = - 0.01 | r = 0.06 | r = 0.42** | r = 0.42** | r = 0.38** | r = 0.38** | r = 0.23 | r = 0.22 | r = 0.40** | r = 0.44** | r = 0.6** | r = 0.60** |
| Serum melatonin | r = 0.09 | r = - 0.01 | r = 0.103 | r = - 0.04 | r = 0.07 | r = 0.07 | r = 0.14 | r = 0.09 | r = 0.03 | r = 0.01 | r = - 0.01 | r = 0.04 |
| Saliva cortisol | r = 0.32* | r = 0.12 | r = 0.20 | r = - 0.05 | r = 0.08 | r = 0.08 | r = 0.05 | r = - 0.03 | r = - 0.11 | r = - 0.13 | r = 0.13 | r = 0.01 |

*Spearman correlation test, significant at $P \leq 0.05$

** $P \leq 0.001$

and meta-analyses conducted on Ramadan fasting, which reported a significant increase in HDL [41] accompanied with a significant reduction in LDL especially in male subjects [42].

During Ramadan, the usual circadian pattern of eating shifts suddenly, resulting in a period of concentrated caloric intake during nighttime hours. The increased caloric intake at night may disrupt the circadian rhythm of metabolic processes during fasting [43]. The control of appetite is governed by leptin, the 'satiety hormone' [44], and ghrelin, the 'hunger hormone' [45], while sleep is governed by cortisol and melatonin [46].

Research interest in leptin and ghrelin has peaked in recent years, with many groups working to understand better the biological functions and mechanisms of these peptide hormones. Ghrelin and leptin were originally believed to primarily act as regulators of feeding and appetite; they have also been shown to play important roles in regulating the functions of other biological systems. At a systemic level, ghrelin and leptin act in concert with a plethora of other signaling molecules.

Leptin was the first adipocytokine to be discovered; it plays a fundamental role in the regulation of appetite and body weight by working as a suppressor of hunger to reduce food intake. Plasma levels of leptin are highly associated with BMI and the degree of adiposity. Leptin concentrations vary in response to changes in dietary caloric intake, with a remarkable decrease in levels during fasting and an increase in obese states. Leptin levels fluctuate over 24-hour cycles, as it is secreted in rhythmic pulses. Leptin levels also exhibit sexual dimorphism, with females having higher leptin levels than males, as leptin synthesis is stimulated by estrogen and inhibited by testosterone [47].

Leptin is an anorexigenic adipokine whose effects oppose those of ghrelin; leptin is mostly produced and released by adipose tissue in concentrations proportional to fat mass, and its circulating concentration declines rapidly in response to reduced body fat levels and food intake, providing a dynamic measure of the size of fat storage compartments and acute changes in energy balance [48]. Leptin's primary metabolic function is to inhibit food intake and stimulate energy expenditure via receptor-mediated actions in the hypothalamus [49]. Serum leptin was assessed in this study and we observed that plasma leptin levels were positively correlated with sex, BMI, hip circumference, and body fat percentage; these findings are in concordance with those reported by Jequier [49].

Ghrelin is an orexigenic peptide produced by specialized endocrine cells of the stomach [50], and circulating ghrelin levels increase before meals, following food deprivation, and after certain forms of weight loss to stimulate the desire for food intake; thus, increased ghrelin levels can be expected in conditions of fasting [51]. The type of dietary intake could also change circulating ghrelin levels, which are reduced following increased carbohydrate and lipid intake [52], and by diets rich in proteins or amino acids [53]. In this study, circulating ghrelin was remarkably reduced at the end of Ramadan DIF. The decrease in circulating ghrelin levels could be attributed to changes in dietary habits during Ramadan, which are richer in fat and carbohydrates. This is consistent with what was observed in this study, where a significant increase in total sugars was observed, with a clear, but non-significant, trend toward the increased intake of total carbohydrates and unsaturated fats (Table 2).

Previous studies have reported a significant reduction in ghrelin levels in six healthy volunteers undergoing fasting for 33 consecutive hours [54]. Moreover, another study demonstrated that neither a 72-hour fasting-induced reduction in leptin levels nor administration of r-metHuLeptin in physiological or high pharmacological doses changed ghrelin levels in human subjects during monitoring periods lasting several hours to a few days [55]. In the current study, ghrelin levels did not increase compared to baseline measures, despite a reduction in leptin levels. These results are consistent with the findings of a previous study, which assessed ghrelin and leptin levels during Ramadan fasting and demonstrated that ghrelin levels did not

increase during fasting [56]. The decrease in plasma leptin levels was not accompanied by an increase in ghrelin levels [56]. It is worth mentioning that BMI in the two previous studies that were discussed and compared to our data (i.e., Chan et al [55] and Alzoghaibi et al. [56]) was normal, 22.3, and 23.7, respectively.

The current study also demonstrated a reduction in plasma melatonin levels in overweight and obese patients participating in DIF during Ramadan. These results are in agreement with previous studies that observed a reduction in melatonin levels, even during short-term experimental fasting [16]. The exact cause of the reduction in melatonin levels is not known; however, some groups have theorized that a reduction in glucose supply during fasting may cause a reduction in melatonin production [57]. Another theory suggested that a decrease in tryptophan may lead to a decrease in melatonin production because tryptophan is an essential component in melatonin production. Nevertheless, this theory is unlikely because Ramadan DIF is not expected to cause a decrease in tryptophan levels [58].

In this study, however, the current data showed no changes in salivary cortisol levels during fasting compared to baseline. Previous studies on Ramadan fasting and cortisol levels exhibited contradicting results [59, 60], which may be ascribed to the different time points and along with different eating and sleep patterns of the fasting participants. Theoretically, one would expect fasting to result in an elevation of cortisol levels for energy mobilization; however, a previous study that assessed plasma cortisol levels in three groups of healthy subjects after fasting for eight to eleven hours reported findings similar to ours [61]. At the end of the fasting period, the investigators randomized participants to 1) glucose load followed by stress; 2) water load followed by stress; or 3) glucose load with no stress. Although there was no difference in plasma cortisol levels at baseline between the three groups, fasting participants treated with glucose demonstrated a clear-cut increase in cortisol response to stress; however, fasting subjects preloaded with water after fasting failed to respond to the psychosocial stressor with a significant cortisol spike [61].

Age had no significant effect on serum ghrelin, melatonin, or leptin levels before or after DIF during Ramadan. Most reports have shown strong associations between serum or plasma ghrelin levels and obesity and disorders of lipid metabolism [62–64]; in these cases, ghrelin levels are reduced due to increased calorie intake in patients with obesity [62], though no association was observed in the present paper.

Some reports suggest an association of ghrelin level in serum. Zhang and colleagues [65] found that plasma ghrelin levels were negatively correlated with body weight, BMI, and total cholesterol levels, and Park et al. [66] reported that fasting plasma ghrelin levels were negatively correlated with body weight, BMI, body fat percentage, waist and hip circumferences, and triglyceride levels, but were positively correlated with HDL levels. In the present study, plasma ghrelin levels were only negatively correlated with waist circumference and positively correlated with male sex. This variability in the relationship between ghrelin hormone levels and other obesity-related parameters could be attributed to variations in sample size, types of food consumed, and different methods of assessments.

One of the limitations of the current work is that it did not control for some confounding factors that may have affected the measured serum parameters, such as sleep duration and timing, total energy expenditure, and light exposure. Moreover, only one sample was collected at each of the two-time points to measure hormone levels; which are not representative of the changes that occur over a typical 24-hour cycle, especially for melatonin which secreted by night and it is normally very low by day. Nevertheless, we wanted to assess the effect of DIF on daytime levels of the above hormones, as this population has not been previously assessed. Thus, it is recommended to take the measurements of hormones several times during the 24 hours and at the end of each week in future studies to overcome the limitation. Another

limitation of the study is that we did not rule out sleep disorders such as obstructive sleep apnea (OSA). Nevertheless, the mean age of the studied group is younger than the classical age of OSA patients, and the group was overweight, not obese. Even for obese ones, they were not massively obese which are more prone to develop OSA. Finally, the external validity was limited due to sample size, and Spearman's correlation does not take into consideration the confounders.

## Conclusion

DIF during Ramadan significantly altered serum levels of ghrelin, melatonin, and serum leptin. Further, male sex and anthropometric variables were the most impacting factors on the tested four hormones. Further studies are needed to assess DIF's impact on the circadian rhythmicity of overweight and obese fasting people.

## Acknowledgments

Authors express their deep thanks to the volunteers for their commitment and enthusiasm during research execution. Thanks are due to Ms. Arwa Fawzan, Ms. Alzahraa Alchaar, and Ms. Yara Hasibi for their significant assistance in data collection.

## Author Contributions

**Conceptualization:** Mohamed Madkour, Mo'ez Al-Islam Faris.

**Data curation:** Mohamed Madkour, Fatima Alhasan, Mo'ez Al-Islam Faris.

**Formal analysis:** Natheer Al-Rawi, Mohamed Madkour, Haitham Jahrami, Dana Salahat.

**Funding acquisition:** Mo'ez Al-Islam Faris.

**Investigation:** Haitham Jahrami, Dana Salahat, Fatima Alhasan, Mo'ez Al-Islam Faris.

**Methodology:** Natheer Al-Rawi, Haitham Jahrami.

**Project administration:** Haitham Jahrami, Mo'ez Al-Islam Faris.

**Resources:** Natheer Al-Rawi, Mo'ez Al-Islam Faris.

**Software:** Mo'ez Al-Islam Faris.

**Supervision:** Mo'ez Al-Islam Faris.

**Validation:** Haitham Jahrami, Ahmed BaHammam, Mo'ez Al-Islam Faris.

**Visualization:** Haitham Jahrami.

**Writing – original draft:** Natheer Al-Rawi, Ahmed BaHammam, Mo'ez Al-Islam Faris.

**Writing – review & editing:** Ahmed BaHammam, Mo'ez Al-Islam Faris.

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
