## [Decision Letter · Decision Letter 0]

15 Jun 2020

PONE-D-20-01785

Effect of diurnal intermittent fasting during Ramadan on ghrelin, leptin, melatonin, and cortisol levels among overweight and obese subjects: A prospective observational study

PLOS ONE

Dear Dr. MoezAlIslam E Faris

Thank you for submitting your manuscript to PLOS ONE. After careful consideration, we feel that it has merit but does not fully meet PLOS ONE’s publication criteria as it currently stands. Therefore, we invite you to submit a revised version of the manuscript that addresses the points raised during the review process.

Based on the comments of the expert reviews , kindly do the revision of the manuscript and resubmit to this journal . 

We look forward to receiving your revised manuscript.

Kind regards,

Nayanatara Arun Kumar

Academic Editor

PLOS ONE

Journal Requirements:

2. lease ensure that your related article ( Madkour, Mohamed I., et al. "Effect of Ramadan diurnal fasting on visceral adiposity and serum adipokines in overweight and obese individuals." Diabetes research and clinical practice 153 (2019): 166-175.) is adequately mentioned in the present submission, and the rationale of these separate analyses is clearly discussed  (for more information on PLOS ONE criteria on related manuscripts , please see http://journals.plos.org/plosone/s/submission-guidelines#loc-related-manuscripts ) .

'Declaration of Competing Interest: None'

5. Please amend the manuscript submission data (via Edit Submission) to include author Fatima Alhasan.

6. Please amend your authorship list in your manuscript file to include author Fatima Alali.

Reviewers' comments:

Reviewer's Responses to Questions

**Comments to the Author**

1. Is the manuscript technically sound, and do the data support the conclusions?

Reviewer #1: Partly

Reviewer #2: Yes

Reviewer #3: Yes

Reviewer #4: Yes

Reviewer #5: Yes

Reviewer #6: Partly

2. Has the statistical analysis been performed appropriately and rigorously? 

Reviewer #1: No

Reviewer #2: Yes

Reviewer #3: No

Reviewer #4: I Don't Know

Reviewer #5: No

Reviewer #6: I Don't Know

3. Have the authors made all data underlying the findings in their manuscript fully available?

Reviewer #1: Yes

Reviewer #2: Yes

Reviewer #3: Yes

Reviewer #4: Yes

Reviewer #5: Yes

Reviewer #6: Yes

4. Is the manuscript presented in an intelligible fashion and written in standard English?

Reviewer #1: Yes

Reviewer #2: Yes

Reviewer #3: Yes

Reviewer #4: No

Reviewer #5: Yes

Reviewer #6: Yes

5. Review Comments to the Author

Reviewer #1: This study measured the effect of diurnal intermittent fasting (DIF) during Ramadan on ghrelin, leptin, melatonin, and cortisol levels among overweight and obese subjects.

The study examined hormone levels (salvia/serum), weight, body type in 57 individuals over a 28 day period. Data points for each individual were collected one week before and after the period of DIF.

The authors found that DIF during Ramadan significantly altered serum levels of ghrelin, melatonin, and serum leptin, and then concluded that these altered hormone levels may impact the circadian rhythms of overweight and obese fasting people.

Major point:

*The authors have identified 57 individuals and assessed each subject for bodyweight/composition, blood hormonal levels, and more. Considering the variability of each individual and that the authors have before and after data for each subject, why were the measurements the subjects and time points averaged? Comparisons of subjects would provide quite a bit more insight to changes observed. A direct relationship of the caloric intake for each individual would be helpful in identifying changes measured. (see data presentation below)

*It seems that averaging all measurements from the subjects could hide specific results for each subject. If combining, then the data must be normalized to be compared.

*Timescale/points – Did the authors consider more intermittent testing rather than one test at the beginning of the study and one test at the end? Could regular testing within the study show trends/rates or changes earlier than 28 days? What if there are more extreme changes in hormone within one week of DIF initiation that they are missing and by waiting 28 days, those levels have leveled off or become more normal?

*According to the data, subjects appeared to consume the same amount of calories before and during DIF. Did the authors link calorie consumption to each individual? Was there direct monitoring of calories or only self-reporting? Are the caloric intake levels reported characteristic of subjects who are obese? Is It possible to supply individuals the same type of food/water and adjust according caloric needs?

*Controls. Is it possible to perform the same experiments on subjects who are eating the same food but not participating in DIF? Do they also experience the same changes or not?

*Did the authors test for levels of estrogen and testosterone for each individual? They state that levels of these two hormones affect levels of leptin. Therefore, if that’s correct, those levels should be measured at the beginning of the study for each.

*Data presentation:

Tables combining means and SD from subjects provide exact data, but visualization of the changes and significance of the studies would be better displayed as histograms or even better, scatter plots where each data point can be viewed in relation to the mean.

I believe that this work is extremely interesting, not only to dieticians but also to the public at large.

Reviewer #2: I appreciate this invitation to review manuscript by MoezAlIslam E Faris et al. entitled: “Effect of diurnal intermittent fasting during Ramadan on ghrelin, leptin, melatonin, and cortisol levels among overweight and obese subjects: A prospective observational study" submitted to PLOS ONE.

MoezAlIslam E Faris et al. investigated the effects of diurnal intermittent fasting during Ramadan on ghrelin, leptin, melatonin, and cortisol levels in 57 overweight and obese subjects, as well as to determine how

anthropometric, dietary and lifestyle changes during Ramadan correlate with the hormonal changes.

The article is quite well written, a study design is simple but quite well planned, and results show interesting findings. However, I have also some minor comments:

1. Introduction- the circadian rhythm should be explained briefly, as well as its importance and role in the metabolic/energy balance regulation.

2. Page 3, “In their critical review (…)” the reference is missed.

3. Page 4, citations are missed.

4. Leptin is described at page 12, next is about ghrelin, and then, at page16 again about leptin, please put it in the thematic order.

Reviewer #3: Dear Editor,

the submitted manuscript is very interesting and one of the few studies in the area of diurnal intermittent fasting. It is very well written and all the required information is well reported.

the author has started the abstract and introduction by focusing on relationship between the selected hormones and cicardian rhythm, however the aim of the study is to study the changes in these hormones before after Ramadan. the hormone melatonin and cortisol may be strongly correlated to sleeping pattern but ghrelin and leptin are also strongly related to huger and satiety(which is not reported in the study) . Since the study is intending to see the relationship between these hormones and ramadan and attribute this affect to change in sleeping pattern, the statistical test used of spearmans correlation is very weak as it does not take into consideration the confounders.

Regression analysis could be a stronger statistical test.

the study design as prospective study, however it is a before after or quasi-experimental study design.

since all the 4 hormones are considered as primary objective, did the author take into consideration the bonferroni correction?

the sleeping pattern is represented by total sleeping hours, maybe day and night hours sleeping could also be reported as Ramadan sleeping pattern reduces night sleeping hours and increases day sleeping hours.

Reviewer #4: Although the article is written with clear English, it is not easy to read. The introduction is too long. This is mainly because there are multiple parameters to introduce (circadian rhythm, the hormones, dietary habits etc.). Especially on the second page, the introduction evolves to a discussion.

Study aims to show the effect of Ramadan on serum levels of ghrelin, leptin, melatonin and salivary cortisol in a group of overweight and obese subjects and seek for correlations with lifestyle changes. Subjects were recruited using social media outlets. Were they paid? How would they correspond to the general population? Especially female subjects were clearly less and middle aged. A control group would have shown if the differences attributed to Ramadan in overweight subjects are also present in normal population, or are they different?

For statistical analysis were all data parametric?

Second paragraph of the discussion belongs to introduction. In fact the discussion doesn't discuss the results of this study, but it explains other studies until the paragraph about ghrelin. Authors explain increased ghrelin levels with increased fat and carbohydrate consuming habits in Ramadan however subjects lost weight and their metabolic parameters ameliorated.

What was the point of including cortisol in this study? Cortisol levels are measured accurately at midnight with salivary measurement. Random cortisol measurement may not reflect cortisol levels optimally. Also cortisol is mostly affected from circadian rhythm, authors may want to explain why it did not differ in Ramadan.

Reviewer #5: The study by Al-Rawi and colleagues investigates the effect of Ramadan fasting on ghrelin, leptin, melatonin and cortisol in overweight and obese subjects. As the authors have pointed out, several changes accompany the Ramadan fast, and these include sleeping patterns and circadian rhythms. As such, a weakness of the presented study is the fact that blood (or saliva in the case of cortisol) samples were only taken at one single time point (8-10 hours after previous meal). The study therefore comes up with no information on what changes to these hormones may be at other times of the day.

In more detail:

Introduction-This is fine.

Methods-

"None of the subjects reported having had sleeping problems". Given the fact that obstructive sleep apnoea and sleeping disorders are highly prevalent among the obese, a more objective attempt at screening for sleep apnoea and other sleeping disorders (such as STOP-BANG questionnaire) would have been useful. If this was not done, a discussion of this and its potential implications should be included in the discussion section.

Study Design-

"subjects were instructed not to alter their habitual physical exercise levels before or during Ramadan". In practice this would be very difficult. Was any attempt made at measuring physical activity. If not, please mention and discuss the implications of this in the discussion section.

Anthropometric assessment

Please include statement on accuracy and reference on validation of the bio-impedance method used.

Neck circumference is reported in table 1. No mention of how this was measured is included in Methods section.

Blood sampling and lipid and hormone assays

"immunosorbence". Please correct to "immunosorbent".

Cortisol measurement was from saliva. Please include a statement about sample collection and precautions taken to ensure reliability.

Dietary intake assessment-

"24-hour recall". Please include a statement about reliability of these and also the software used. Reference?

Statistical analysis-

"Two-tailed paired sample t-tests..." Were data normally distributed? Please include a statement on this.

Results:

"BMI 29.89". Unit? Also inappropriate decimals. Suggest change to 29.9 kg/m2.

"89.48%". Suggest change to "89.9%".

Page 11. Line 8. "The male sex...". Suggest change to "Male sex...".

Plasma levels of lipids and hormones:

Were the reported changes in hormones independent of weight (and also of sleep duration)?

Discussion:

This section needs to discuss the study findings in more depth. Comparison with previous studies should be more comprehensive. In the case of cortisol for example, some important papers on changes in cortisol circadian rhythm have not been mentioned. In the case of leptin, as mentioned above, statistical analysis should look into whether changes in leptin with Ramadan fasting are weight and body fat independent. The authors have specifically investigated the obese and the overweight, there is no discussion of how similar, or different the findings of this study are from those in normal weight subjects.

Tables:

Two decimals have been reported for all parameters in the tables. Please consider whether this is appropriate. Oxygen saturation is almost identical for T1 & T2; yet the difference is highly significant statistically (p<0.001). Please check this again. Moreover, what was the reason for checking Oxygen saturation? What are the theoretical grounds for expecting a difference?

Highly significant changes in LDL and HDL cholesterol have been reported in table 3, but not discussed at all in the discussion section.

References:

Please see note above on discussion.

In summary:

Good effort by the authors. For a study of hormones affected by circadian rhythms, single time points comparisons in pre- and post-Ramadan periods (as opposed to multiple, or at least two-point comparisons) is the most important weakness. This has been acknowledged by the authors. Further in-depth analysis of the findings of the study are suggested.

Reviewer #6: PONE-D-20-01785

Title: Effect of diurnal intermittent fasting during Ramadan on ghrelin, leptin, melatonin, and cortisol levels among overweight and obese subjects: A prospective observational study

1- Abstract

• organize by adding background, methods, results, conclusion

• In methods, change this sentence « at fixed times of the day » by another one giving the hour of sampling

• Give details in the results paragraph: Give the time about « serum levels of ghrelin, melatonin, and leptin significantly»

• Give results on sleep (result of Stanford Health Care Sleep Questionnaire)

• The conclusion at the end of the abstract : “which may impact the circadian rhythmicity of overweight and obese fasting people” is not accurate because authors did not study circadian rhythm since they have made the assessment at one time point (From 11 am To 1 pm)

2- Introduction

• Replace reference 3 by another one more general about the relation between food and circadian clock.

• Replace reference 5 which is not about Ramadan fasting, give a review or original article on Ramadan and chronobiology

• Replace reference 7 with another on dietary habits and Ramadan, done in different Islamic countries

• Reference 8 is not complete, add ref of an original article on sleep and Ramadan

Paragraph 2: make sentences shorter

Paragraph 5 need to be change to make it more clear, and more organized

3- Methods

• First paragraph line 10 : replace the verb eliminated

• Second paragraph : give details about criteria : regular sleep/wake schedule

• In the paragraph entitled anthropometric assessment, authors written that subjects were assessed in the late morning (11:00 am-1:00 pm) after fasting for eight to ten hours. This means that food intake at the previous night could be at 1 or 3 am, which means that sleep in control days was very delayed and not “regular” as mentioned in including criteria

4- Results

• Table 4 : All these correlations just make the reading difficult, they are not all necessary; make it shorter in term of data

• Discussion

The discussion is too long and focused on the result of others study. Authors need to discuss more their own results, particularly some limitations which will allow the reader to make the right conclusions

• The Most important limitation was choosing one point at 11 am or 1 pm to determine the effect of Ramadan on melatonin and cortisol. The previous studies showed that the effect of Ramadan on these two hormones was an increase by night and a decrease in the morning. So the experimental design of this study will not allow to know the real effect of Ramadan on melatonin or cortisol. Thus the authors must always gave the time of saliva intake and they need to discuss this limitation

• Authors did not study the circadian rhythm, so in the first paragraph line 6, authors could not write: “circadian rhythms such as ghrelin, leptin, and melatonin were significantly reduced at the end of the fasting. To do such a study authors need several sampling time a cross the 24H.

• About change in melatonin, authors compared Ramadan fasting and experimental one and cited the ref 37. This reference is not about experimental fasting and could not be used to argue this comparison.

• This study did not gave a reliable result about melatonin; since this hormone is secreted by night and it is normally very low by day. So we don’t expect reliable result if we study it with one time point in the morning

• I recommend to the authors to reduce the part reserved in the discussion to cortisol and melatonin. Specially that this study did not use the appropriate time to measure these 2 hormones

• Paragraph 4 line 4 : ref 31 not appropriate because not related to intermittent fasting

• The conclusion must be changed. It did not reflect the result. For example the result showed significant correlations between some variables

6. PLOS authors have the option to publish the peer review history of their article (what does this mean?). If published, this will include your full peer review and any attached files.

Reviewer #1: No

Reviewer #2: No

Reviewer #3: No

Reviewer #4: No

Reviewer #5: No

Reviewer #6: No

---

## [Author Response · Author response to Decision Letter 0]

9 Jul 2020

Authors Responses to Reviewers' Comments

PONE-D-20-01785

Effect of diurnal intermittent fasting during Ramadan on ghrelin, leptin, melatonin, and cortisol levels among overweight and obese subjects: A prospective observational study

Reviewers' evaluation 1: Has the statistical analysis been performed appropriately and rigorously? Three out of six reviewers indicated for the inappropriate and non-rigorous statistical analysis performed.

Response: The frequency distribution (histogram) and boxplot were used for checking normality visually. 

Parametric tests are more robust than non-parametric tests in all cases. https://www.ncbi.nlm.nih.gov/pmc/articles/PMC3668290/

Spearman's correlation determines the strength and direction of the monotonic relationship between two variables rather than the strength and direction of the linear relationship between two variables, which is what Pearson's correlation determines.

Reviewers' evaluation 2: Did the authors made all data underlying the findings in their manuscript fully available?.

Response: All the reviewers indicated that the authors made all data underlying the findings in their manuscript fully available. Thanks a lot. 

Reviewers' evaluation 3: Is the manuscript presented in an intelligible fashion and written in standard English?

Response: Five out of six reviewers indicated that the manuscript presented in an intelligible fashion and written in standard English. Thank you.

Rebuttal reviewers' Comments:

Reviewer #1: 

This study measured the effect of diurnal intermittent fasting (DIF) during Ramadan on ghrelin, leptin, melatonin, and cortisol levels among overweight and obese subjects. The study examined hormone levels (salvia/serum), weight, body type in 57 individuals over 28 days. Data points for each individual were collected one week before and after the period of DIF. The authors found that DIF during Ramadan significantly altered serum levels of ghrelin, melatonin, and serum leptin, and then concluded that these altered hormone levels may impact the circadian rhythms of overweight and obese fasting people.

Major point:

Comment 1: The authors have identified 57 individuals and assessed each subject for body weight/composition, blood hormonal levels, and more. Considering the variability of each individual and that the authors have before and after data for each subject, why were the measurements the subjects and time points averaged? Comparisons of subjects would provide quite a bit more insight into the changes observed. A direct relationship of the caloric intake for each individual would help identify changes measured. (see data presentation below)

Response: Excellent point. However, the average/aggregate score is akin to ITT/ITH analysis analogy. This also allows examining heterogeneity of the change. This approach was used in previous studies reviewed and allows cross-discussion. 

Comment 2: It seems that averaging all measurements from the subjects could hide specific results for each subject. If combining, then the data must be normalized to be compared.

Response: True, Z score will provide better estimates. However, interpretation will be impossible due to a lack of comparative studies presenting normalization. Min-max normalization was one of the options too to normalize data. But denied for departures from previous studies carrying similar results. Many machine learning algorithms may help solving the problem above, KNIME analytics will be studied in future studies. 

*Timescale/points 

Comment 3: Did the authors consider more intermittent testing rather than one test at the beginning of the study and one test at the end? 

Response: No. Will be applied in future studies. Thanks for this valuable suggestion. 

Comment 4: Could regular testing within the study show trends/rates or changes earlier than 28 days? 

Response: A study that measured leptin and ghrelin levels at baseline and after 1 and 2 weeks of fasting in a small sample of volunteers (n=8) reported no significant differences in the level of both hormones in week 1 and week 2 (Alzoghaibi et al. Doi: 10.1371/journal.pone.0092214). Another study measure melatonin levels at bases line, and after 1 and 3 weeks of intermittent fasting in a small sample of volunteers (n=8) reported no significant differences in the melatonin levels in week 1 and week 3 (BaHammam. Sleep Biol Rhythm 2004). A third study measured melatonin levels at the baseline and after 1 and 2 weeks of intermittent fasting in a sample of 8 healthy volunteers and reported no significant differences in melatonin levels in week 1 and 3 (Almeneessier et al. Doi: 10.4103/atm.ATM_15_17)

Comment 5: What if there are more extreme changes in hormone within one week of DIF initiation that they are missing and by waiting 28 days, those levels have leveled off or become more normal?

Response: We are looking at the cumulative effect of the fasting month; especially, we are dealing with normal subjects, not diseased patients. Thus no fear of any adverse impact if the level of the hormone increased to the extreme during the month. Nonetheless, we cannot rule out that possibility; however, based on the data from small studies that were discussed in the above response, this possibility is less likely. Nevertheless, the fact that measurements were only done at the end of the month was added as a limitation of the study. Moreover, it was recommended in the revised draft to take the measurements of hormones at the end of each week in future studies. See page 18, lines 255-461.

Comment 6: According to the data, subjects appeared to consume the same amount of calories before and during DIF. Did the authors link calorie consumption to each individual?

Response: We have measured the individual dietary intakes and then averaged the total intake. Thus intake is not linked to each individual separately. Several reports indicated the lack of significant change of dietary intakes before and during Ramadan fasting. 

- https://www.sciencedirect.com/science/article/pii/S0271531712001820

- https://www.sciencedirect.com/science/article/pii/S0168822719302050

Comment 7: Was there direct monitoring of calories or only self-reporting? 

Response: It was self-reporting, free-living subjects without monitoring. 

Comment 8: Are the caloric intake levels reported characteristics of subjects who are obese?

Response: The caloric intake was reported for both overweight and obese subjects. We found that the dietary intakes were nearly balanced and within the AMDR ratios. A matter that leads to speculation that obesity was mostly a result of physical inactivity and the lack of purposeful programmed physical exercise lifestyle (as reported in the manuscript) rather than the excessive caloric intake. 

Comment 9: Is it possible to supply individuals with the same type of food/water and adjust according to caloric needs?

Response: No, as the study was completed during the last Ramadan month. This could be recommended in future research. Highly valuable point. Thanks a lot.

Comment 10: Is it possible to perform the same experiments on subjects who are eating the same food but not participating in DIF?

Response: The current research is not experimental, but rather an observational study in the free-living environment. Theoretically, yes. However, during Ramadan, it is very difficult to find non-fasting people from the same ethnic, cultural, and religious background, as RDIF is a religious type of intermittent fasting. 

Comment 11: Do they also experience the same changes or not?

Response: Expected to have different changes, but this cannot be confirmed without real testing. 

Comment 12: Did the authors test for levels of estrogen and testosterone for each individual? They state that levels of these two hormones affect levels of leptin. Therefore, if that’s correct, those levels should be measured at the beginning of the study for each.

Response: No, didn't. Although many studies stated the presence of a relationship between leptin and sex hormones, another research indicates no association between them (Alexander, Cochran, et al. 2010, Xing, Liu, et al. 2015). So, it's contradictory findings. Also, these sex hormones altered, especially in females throughout the whole month, for deep looking at them; at least, a serum sample needs to be collected weekly from the participant. This could be applied in future research. 

Comment 13: *Data presentation: Tables combining means and SD from subjects provide exact data, but visualization of the changes and significance of the studies would be better displayed as histograms or even better, scatter plots where each data point can be viewed in relation to the mean.

Response: See the reply in Comment 2. 

Comment 14: I believe that this work is extremely interesting, not only to dieticians but also to the public at large.

Response: Thanks a lot for your compliment 

Reviewer #2: 

Comment 15: I appreciate this invitation to review the manuscript by MoezAlIslam E Faris et al. entitled: “Effect of diurnal intermittent fasting during Ramadan on ghrelin, leptin, melatonin, and cortisol levels among overweight and obese subjects: A prospective observational study" submitted to PLOS ONE.

MoezAlIslam E Faris et al. investigated the effects of diurnal intermittent fasting during Ramadan on ghrelin, leptin, melatonin, and cortisol levels in 57 overweight and obese subjects, as well as to determine how anthropometric, dietary and lifestyle changes during Ramadan correlate with the hormonal changes.

The article is quite well written, the study design is simple but quite well planned, and the results show interesting findings. 

Response: Thanks for your compliment

However, I have also some minor 

Comments:

Comment 16: Introduction- the circadian rhythm should be explained briefly, as well as its importance and role in the metabolic/energy balance regulation.

Response: Added. See page 3, Lines 80-82. 

Comment 17: Page 3, “In their critical review (…)” the reference is missed.

Response: Added. This paragraph was transferred to the discussion section based on one reviewer's comment. See page 14, line 344. (Ref [3]).

Comment 18: Page 4, citations are missed.

Response: Two reference added [30, 31]. Page 4, line 128. 

Comment 19: Leptin is described on page 12, next is about ghrelin, and then, at page16 again about leptin, please put it in the thematic order.

Response: The paragraph about leptin is transferred from page 16 to page 13 in continuum with leptin paragraph. See pages 15-16, lines 379-387. 

Reviewer #3: 

Comment 20: Dear Editor, the submitted manuscript is very interesting and one of the few studies in the area of diurnal intermittent fasting. It is very well written and all the required information is well reported. 

Response: Thanks a lot for your compliment 

Comment 21: The author has started the abstract and introduction by focusing on the relationship between the selected hormones and circadian rhythm; however the study aims to study the changes in these hormones before after Ramadan. The hormone melatonin and cortisol may be strongly correlated to the sleeping pattern but ghrelin and leptin are also strongly related to hunger and satiety (which is not reported in the study). 

Response: Addressed. See page 15, lines 360-362. 

Comment 22: Since the study is intending to see the relationship between these hormones and Ramadan and attribute, this effect to change in sleeping pattern, the statistical test used of spearman's correlation is very weak as it does not take into consideration the confounders.

Response: Agree, this will be added to limitations. But to justify Spearman's correlation determines the strength and direction of the monotonic relationship between two variables rather than the strength and direction of the linear relationship between two variables, which is what Pearson's correlation determines.

Comment 26: Regression analysis could be a stronger statistical test.

Response: Due to the limited sample size, the analysis is not powered for multiple regression. High col-linearity is observed. 

Comment 27: The study design as prospective study, however it is a before after or quasi-experimental study design.

Response: The study design is observational prospective design rather than before and after or quasi-experimental. The latter are experimental designs, while the current is observational. See: "These, Matthew S. "Observational and interventional study design types; an overview." Biochemia Medica, 24.2 (2014): 199-210", and" Sedgwick, Philip. "Before and after study designs." BMJ 349 (2014)". 

Further, Describing Ramadan fasting research as prospective observational is well established and can be found in a long list of published studies, for example:

- Nematy, Mohsen, et al. "Effects of Ramadan fasting on cardiovascular risk factors: a prospective observational study." Nutrition Journal 11.1 (2012): 69.

https://pubmed.ncbi.nlm.nih.gov/22963582/

- Salahuddin, M., et al. "Effect of Ramadan fasting on body weight,(BP) and biochemical parameters in middle-aged hypertensive subjects: an observational trial." Journal of clinical and diagnostic research: JCDR 8.3 (2014): 16.

https://www.ncbi.nlm.nih.gov/pmc/articles/PMC4003623/

- Akaberi, Arash, et al. "Does fasting in Ramadan ameliorate Lipid profile? A prospective observational study." Pakistan journal of medical sciences 30.4 (2014): 708.

https://www.ncbi.nlm.nih.gov/pmc/articles/PMC4121682/

Comment 28: Since all the 4 hormones are considered as primary objective, did the author take into consideration the Bonferroni correction?

Response: We performed Tukey MTC but not reported due to a lack of additional value. 

Comment 29: The sleeping pattern is represented by total sleeping hours, maybe day and night hours sleeping could also be reported as Ramadan sleeping pattern reduces night sleeping hours and increases day sleeping hours. 

Response: You are right, but we have combined both as the total sleep duration, which is the most important factor. This could be considered in future research. 

We agree with the reviewer; lifestyle changes during Ramadan may influence the sleep pattern of fasting performers. Sleep pattern was not assessed in this study. Therefore, we have added this as a limitation in the study and advised for assessing the relationship between hormonal levels and sleep patterns in future studies.

Reviewer #4: 

Comment 30: Although the article is written with clear English, it is not easy to read. The introduction is too long. This is mainly because there are multiple parameters to introduce (circadian rhythm, the hormones, dietary habits, etc.). Especially on the second page, the introduction evolves to a discussion.

Response: The third paragraph of the second page of the introduction is transferred to the discussion section. Page 14, lines 344-349. 

Comment 31: Study aims to show the effect of Ramadan on serum levels of ghrelin, leptin, melatonin, and salivary cortisol in a group of overweight and obese subjects and seek correlations with lifestyle changes. Subjects were recruited using social media outlets. Were they paid? 

Response: No monetary or non-monetary incentives were given to the participants. The participation was completely voluntary. See page 6, lines 142-144. 

Comment 32: How would they correspond to the general population? Especially female subjects were less and middle aged.

Response: We agree, the external validity is limited due to sample size, this was added to limitations. See page 19, lines 465-466. 

Comment 33: A control group would have shown if the differences attributed to Ramadan in overweight subjects are also present in normal population, or are they different?

Response: Theoretically, right; however, during Ramadan, it is very difficult to find non-fasting people from the same ethnic, cultural, and religious backgrounds, as RDIF is a religious type of intermittent fasting practiced by almost all Muslim people.

Comment 34: For statistical analysis were all data parametric?

Response: Parametric tests are more robust than non-parametric tests in all cases. https://www.ncbi.nlm.nih.gov/pmc/articles/PMC3668290/

The frequency distribution (histogram) and boxplot, were used for checking normality visually. 

Comment 35: Second paragraph of the discussion belongs to the introduction. The discussion doesn't discuss the results of this study, but it explains other studies until the paragraph about ghrelin. Authors explain increased ghrelin levels with increased fat and carbohydrate consuming habits in Ramadan however subjects lost weight and their metabolic parameters ameliorated.

Response: The second paragraph of the discussion was deleted and transferred to the introduction. Further, the original paragraph 2 has been partly deleted, and this part was combined with paragraph 3: “Research interest in leptin and ghrelin has peaked in recent years, with many groups working to understand better the biological functions and mechanisms of these peptide hormones. Ghrelin and leptin were originally believed to primarily act as regulators of feeding and appetite; they have also been shown to play important roles in regulating the functions of other biological systems. At a systemic level, ghrelin and leptin act in concert with a plethora of other signaling molecules.” See page 15, lines 363-368. 

Comment 36: What was the point of including cortisol in this study? 

Response: Cortisol is one of the most important four hormones (melatonin, ghrelin, and leptin) that are directly involved in the circadian rhythms. See: "Schmid, Dagmar A., et al. "Changes of sleep architecture, spectral composition of sleep EEG, the nocturnal secretion of cortisol, ACTH, GH, prolactin, melatonin, ghrelin, and leptin, and the DEX-CRH test in depressed patients during treatment with mirtazapine." Neuropsychopharmacology 31.4 (2006): 832-844", and the most commonly tested hormones in this field of study: "Allison, Kelly C., et al. "Neuroendocrine profiles associated with energy intake, sleep, and stress in the night eating syndrome." The Journal of Clinical Endocrinology & Metabolism 90.11 (2005): 6214-6217. Further, cortisol has been linked to chronic stress or endogenous hypercortisolism and associated with insulin resistance. This hormone is not well studied during Ramadan intermittent fasting. Therefore, we included cortisol as one of the hormones to be measured in this study. 

Comment 37: Cortisol levels are measured accurately at midnight with salivary measurement. Random cortisol measurement may not reflect cortisol levels optimally. Also, cortisol is mostly affected from circadian rhythm; authors may want to explain why it did not differ in Ramadan.

Response: We agree with the reviewer; therefore, we added this interpretation in the discussion: “Theoretically, one would expect that fasting will result in an elevation of cortisol levels for energy mobilization; however, a previous study that assessed plasma cortisol levels in three groups of healthy subjects after fasting for eight to eleven hours reported findings similar to ours (Kirschbaum, et al., 1997). At the end of the fasting period, the investigators randomized participants to 1) glucose load followed by stress; 2) water load followed by stress; or 3) glucose load with no stress. Although there was no difference in plasma cortisol levels at baseline between the three groups, fasting participants treated with glucose demonstrated a clear-cut increase in cortisol response to stress; however, fasting subjects preloaded with water after fasting failed to respond to the psychosocial stressor with a significant cortisol spike (Kirschbaum, et al., 1997). See pages 17-18, lines 427-436. 

Reviewer #5:

 The study by Al-Rawi and colleagues investigates the effect of Ramadan fasting on ghrelin, leptin, melatonin, and cortisol in overweight and obese subjects. As the authors have pointed out, several changes accompany the Ramadan fast, and these include sleeping patterns and circadian rhythms. 

Comment 38: As such, a weakness of the presented study is the fact that blood (or saliva in the case of cortisol) samples were only taken at one single time point (8-10 hours after a previous meal). The study, therefore, comes up with no information on what changes to these hormones may be at other times of the day.

Response: We have added this as a limitation of the study: “Moreover, only one sample was collected at each of the two-time points to measure hormone levels, which are not representative of the changes that occur over a typical 24-hour cycle. Nevertheless, we wanted to assess the effect of DIF on daytime levels of the above hormones, as this population has not been previously assessed.” See pages 18-19, lines 455-461.

In more detail: 

Comment 39: Introduction-This is fine. 

Response: Thanks

Comment 40: Methods-"None of the subjects reported having had sleeping problems". Given the fact that obstructive sleep apnea and sleeping disorders are highly prevalent among the obese, a more objective attempt at screening for sleep apnea and other sleeping disorders (such as STOP-BANG questionnaire) would have been useful. If this was not done, a discussion of this and its potential implications should be included in the discussion section. 

Response: Obstructive sleep apnea and sleep-related breathing disorders are more common in obese people (obese grade III), which is not the case with the study participants who were mostly overweight and obese grade I and the least grade II. 

We agree with this possibility, although the mean age of the studied group is younger than the classical age of OSA. Additionally, the group was mostly overweight, not obese, and even for obese people, they were not in a massive state. Nevertheless, the following has been added to the discussion: “Another limitation of the study is that we did not rule out sleep disorders such as obstructive sleep apnea. Nevertheless, the mean age of the studied group is younger than the classical age of obstructive sleep apnea patients, and the group was overweight, not obese.” See page 19, lines 461-466.

Study Design-

Comment 41: "subjects were instructed not to alter their habitual physical exercise levels before or during Ramadan". In practice, this would be very difficult. Was any attempt made at measuring physical activity. If not, please mention and discuss the implications of this in the discussion section.

Response: Physical activity was already assessed and reported in a single paragraph, and discussed as well. See page lines, and in the discussion, see page 10, lines 236-249. 

Anthropometric assessment 

Comment 42: Please include statement on accuracy and reference on validation of the bio-impedance method used. 

Response: Validation and high accuracy of DSM-BIA in the quantification of total body and segmental body composition had been reported elsewhere (Ling, de Craen, et al. 2011, De Rui, Veronese, et al. 2017). See page 8, lines 186-188.

Comment 43: Neck circumference is reported in table 1. No mention of how this was measured is included in the Methods section. 

Response: Neck circumference was measured in the midway of the neck, between the mid-cervical spine and mid anterior neck, to within 1 mm, with non-stretchable plastic tape with the subjects standing upright. In men with a laryngeal prominence (Adam's apple), it was measured just below the prominence (Aswathappa, Garg, et al. 2013). See page 8, lines 196-200. 

Comment 44: Blood sampling and lipid and hormone assays "immunosorbence". Please correct to "immunosorbent".

Response: Corrected. Page 8, lines 210; page 24, Abbreviations. 

Comment 45: Cortisol measurement was from saliva. Please include a statement about sample collection and precautions taken to ensure reliability. 

Response: Added. 

Unstimulated saliva was collected and processed according to the Salimetrics manufacture manual. For saliva collection, the fasting donors already came while avoided eating, drinking, and smoking, and were instructed not to use oral hygiene products for at least 1 hour before their hospital visit. Spit samples were collected by asking volunteers to forcefully spit saliva (not sputum) into the sterile plastic saliva collection containers through direct spitting. The collected samples were centrifuged at 1500 gx for 15 min at 4°C within one hour of saliva collection. After centrifuge, the supernatants were immediately separated by sterile pipettes, aliquoted into smaller volumes, and stored at -80°C deep freezer until used for cortisol assay. To avoid hormone degradation, thawed saliva samples were used once. See page 9, lines 218-226. 

Comment 46: Dietary intake assessment-"24-hour recall". Please include a statement about the reliability of these and also the software used. Reference?

Response: Validity and reliability of 24-hour recall in assessing dietary and caloric intake are reported elsewhere (Sun, Roth, et al. 2010, Beaton, Wright, et al. 2018, Foster, Lee, et al. 2019). See page 9, lines 233-234. 

 Comment 47: Statistical analysis-"Two-tailed paired sample t-tests..." Were data normally distributed? Please include a statement on this.

Response: The frequency distribution (histogram) and boxplot were used for checking normality visually. 

Results:

Comment 48: "BMI 29.89". Unit? Also inappropriate decimals. Suggest change to 29.9 kg/m2."89.48%". Suggest change to "89.9%". 

Response: Corrected as suggested. See page 12, lines 290-291. 

Comment 49: Page 11. Line 8. "The male sex...". Suggest change to "Male sex...". 

Response: Corrected as suggested. See page 13, line 322. 

Comment 50: Plasma levels of lipids and hormones: 

Were the reported changes in hormones independent of weight (and also of sleep duration)?

Response: No. 

Discussion:

Comment 51: This section needs to discuss the study findings in more depth. A comparison with previous studies should be more comprehensive. In the case of cortisol, for example, some important papers on changes in cortisol circadian rhythm have not been mentioned.

Response: 

Previous studies on Ramadan fasting and cortisol levels exhibited contradicting results (Al-Hadramy, Zawawi, et al. 1988, Bahijri, Borai, et al. 2013), which may be ascribed to the different time points and along with different eating and sleep patterns of the fasting participants. See page 17, lines 425-427. 

Comment 52: In the case of leptin, as mentioned above, statistical analysis should look into whether changes in leptin with Ramadan fasting are weight and body, fat independent. The authors have specifically investigated the obese and the overweight, there is no discussion of how similar, or different the findings of this study are from those in normal-weight subjects. 

Response: The BMI in the studies that were discussed and compared to our data (i.e., Chan et al and Alzoghaibi et al.) was normal, 22.3, and 23.7, respectively. See page 17, lines 410-412. 

Tables:

Comment 53: Two decimals have been reported for all parameters in the tables. Please consider whether this is appropriate

Response: All values in Tables 1-3 are changed into one decimal.

Comment 54: Oxygen saturation is almost identical for T1 & T2, yet the difference is highly significant statistically (p<0.001). Please check this again. 

Response: Oxygen saturation is removed to minimize confusion data without clinical significance. 

Comment 55: Moreover, what was the reason for checking Oxygen saturation? What are the theoretical grounds for expecting a difference?

Response: Oxygen saturation is removed to minimize confusion data without clinical significance. 

Comment 56: Highly significant changes in LDL and HDL cholesterol have been reported in table 3, but not discussed at all in the discussion section. 

Response: The significant changes reported in HDL and LDL at the end of the fasting month in comparison with the pre-fasting levels are consistent with the vast majority of systematic reviews and meta-analyses conducted on Ramadan fasting, which reported a significant increase in HDL (Faris, Jahrami et al. 2020) accompanied with a significant reduction in LDL especially in male subjects (Kul, Savaş et al. 2014). See page 14, lines 351-355. 

References:

Comment 57: Please see note above on discussion. 

Response: Not clear.

In summary: 

Comment 58: Good effort by the authors. 

Response: Thank you for the compliment

Comment 59: For a study of hormones affected by circadian rhythms, single time points comparisons in pre- and post-Ramadan periods (as opposed to multiple, or at least two-point comparisons) is the most important weakness. This has been acknowledged by the authors.

Response: Thank you. We have addressed that in the limitations. See pages 18-19, lines 455-461.

Comment 60: Further in-depth analysis of the findings of the study are suggested.

Response: What type of test is suggested for in-depth analysis? Authors ready to do any requested test. 

Reviewer #6:

Title: Effect of diurnal intermittent fasting during Ramadan on ghrelin, leptin, melatonin, and cortisol levels among overweight and obese subjects: A prospective observational study

1- Abstract 

Comment 61: Organize by adding background, methods, results, conclusion

Response: Done, inserted. See Abstract, page 2. 

Comment 62: In methods, change this sentence « at fixed times of the day » by another one giving the hour of sampling.

Response: Added (11:00 am-1:00 pm). See page 7, line 179-180. 

Comment 63: Give details in the results paragraph: Give the time about « serum levels of ghrelin, melatonin, and leptin significantly»

Response: Time mentioned as (11:00 am-1:00 pm) before and after Ramadan fasting. See page2, line 67.

Comment 64: Give results on sleep (result of Stanford Health Care Sleep Questionnaire).

Response: Total sleep time as calculated apart from the Stanford Health Care Sleep Questionnaire, along with the presence of sleep problems, were the only parts of the questionnaire used. Total sleep duration is already reported at the top of Table 2. 

Comment 65: The conclusion at the end of the abstract: “which may impact the circadian rhythmicity of overweight and obese fasting people” is not accurate because authors did not study circadian rhythm since they have made the assessment at one time point (From 11 am To 1 pm). 

Response: We agree with the reviewer. The conclusion now reads:

Conclusion: DIF during Ramadan significantly altered serum levels of ghrelin, melatonin, and serum leptin. Further studies are needed to assess DIF's impact on the circadian rhythmicity of overweight and obese fasting people. See page 2, lines 72-75. 

2-Introduction 

Comment 66: Replace reference 3 by another one more general about the relation between food and circadian clock.

Response: Replaced with "Asher, Gad, and Paolo Sassone-Corsi. "Time for food: the intimate interplay between nutrition, metabolism, and the circadian clock." Cell 161.1 (2015): 84-92". and " Challet, Etienne. "The circadian regulation of food intake." Nature Reviews Endocrinology 15.7 (2019): 393-405." See page 3, line 84. 

Comment 67: Replace reference 5 which is not about Ramadan fasting, give a review or original article on Ramadan and chronobiology. 

Response: Reference 5 is replaced with a review article and two original articles. See page 2, line 66, Ref (Faris, Madkour et al. 2019; Madkour, El-Serafi, et al. 2019; BaHammam and Almeneessier 2020, Faris, Jahrami, et al. 2020). See page 3, line 91.

Comment 68: Replace reference 7 with another on dietary habits and Ramadan, done in different Islamic countries.

Response: Replaced with two references, one from Iran and the other from Turkey. During the month of Ramadan, several changes have been reported in dietary habits, meal frequency and timing (Karaagaoglu and Yucecan 2000, Shadman, Poorsoltan, et al. 2014, Almeneessier, BaHammam, et al. 2019). See page 3, line 93. 

Comment 69: Reference 8 is not complete, add ref of an original article on sleep and Ramadan.

Response: Reference 8 is complemented now. Original research on RF and sleep is added. "BaHammam A. Effect of fasting during Ramadan on sleep architecture, daytime sleepiness, and sleep pattern. Sleep Biol Rhythm. 2004; 2".[15,16]. See page 3, line 93. 

Comment 70: Paragraph 2: make sentences shorter

Response: Shortened as suggested. Please see page 2, lines 93-97. 

Comment 71: Paragraph 5 need to be change to make it clearer, and more organized.

Response: Paragraph 5 is transferred to the discussion as suggested by one of the reviewers. As well, the paragraph is rephrased for better clearance. See page 14, line 344-349.

3- Methods 

Comment 72: First paragraph line 10: replace the verb eliminated 

Response: Replaced with "excluding". See page 6, line 149. 

Comment 73: Second paragraph: give details about criteria: regular sleep/wake schedule.

Response: "Criteria of regular sleep/wake schedule involves the lack of erratic sleep patterns, and having a fixed schedule of sleep/awake without continuous interruptions, thus enabling meeting the body needed sleep hours each day, avoid feeling with daytime sleepiness". See page 6, line 155-158.

Comment 74: In the paragraph entitled anthropometric assessment, authors written that subjects were assessed in the late morning (11:00 am-1:00 pm) after fasting for eight to ten hours. This means that food intake at the previous night could be at 1 or 3 am, which means that sleep in control days was very delayed and not “regular” as mentioned in including criteria.

Response: Before RIF, the participant were asked to come fasting at least 8 – 10 hours; and we delay it purposely at the end of RIF to cover the eight hours after suhoor (3-4 am). At least does not merely mean 8-10 exactly, it was longer than that because they were on regular sleep/wake cycle, but this is the minimum time requested from the participants. 

 4- Results 

Comment 75: Table 4: All these correlations just make the reading difficult, they are not all necessary; make it shorter in term of data.

Response: Sleep duration and total caloric intake without significant correlations were removed from the table. 

Discussion

Comment 76: The discussion is too long and focused on the result of others study. Authors need to discuss more their own results, particularly some limitations which will allow the reader to make the right conclusions.

Response: The discussion has been shortened and tailored to the current study results. Please look at the limitations page 18-19, lines 455-466. 

Comment 77: The Most important limitation was choosing one point at 11 am or 1 pm to determine the effect of Ramadan on melatonin and cortisol. The previous studies showed that the effect of Ramadan on these two hormones was an increase by night and a decrease in the morning. So the experimental design of this study will not allow knowing the real effect of Ramadan on melatonin or cortisol. Thus the authors must always gave the time of saliva intake and they need to discuss this limitation

Response: Legitimate point that is added to the study limitation, and suggested to be considered in future research (see page 18-19, lines 455-466). However, the time of saliva collection is already mentioned (11:00 am-1:00 pm). See page 7, line 179-180. 

 Comment 78: Authors did not study the circadian rhythm, so in the first paragraph line 6, authors could not write: “circadian rhythms such as ghrelin, leptin, and melatonin were significantly reduced at the end of the fasting. To do such a study authors need several sampling time a cross the 24H. 

Response: This statement has been corrected in the discussion. It reads now as: "Further, serum levels of the hormones leptin, ghrelin, and melatonin were significantly reduced at the end of the fasting month compared to the pre-fasting state". Page 14, lines 338-339.

Comment 79: About change in melatonin, authors compared Ramadan fasting and experimental one and cited the ref 37. This reference is not about experimental fasting and could not be used to argue this comparison.

Response: Comparison should be with the results of the same design. Our study is observational, not experimental, in nature. Reference 37 (BaHammam. J Sleep Biol Rhythm 2004) was not experimental fasting. It was an observational study on volunteers during Ramadan intermittent fasting. 

Comment 80: This study did not give a reliable result about melatonin; since this hormone is secreted by night and it is normally very low by day. So we don’t expect reliable result if we study it with one time point in the morning.

Response: A limitation of this study is the single measurement of hormones, which was stressed in the limitation section of the paper. See page 18-19, lines 455-466.

 Comment 81: I recommend to the authors to reduce the part reserved in the discussion to cortisol and melatonin. Specially that this study did not use the appropriate time to measure these 2 hormones. 

Response: Paragraphs related to the discussion of cortisol and melatonin are reduced.

Comment 82: Paragraph 4 line 4 : ref 31 not appropriate because not related to intermittent fasting. 

Response: Changed (Muller, Lamberts, et al. 2002). See page 16, line 392. 

Comment 83: The conclusion must be changed. It did not reflect the result. For example the result showed significant correlations between some variables

Response: The conclusion is changed. See page 2, line 72-75, and page 19, lines 470-473. 

Al-Hadramy, M. S., T. H. Zawawi, and S. M. Abdelwahab (1988). "Altered cortisol levels in relation to Ramadan." Eur J Clin Nutr 42(4): 359-362.

Alexander, C., C. J. Cochran, L. Gallicchio, S. R. Miller, J. A. Flaws, and H. Zacur (2010). "Serum leptin levels, hormone levels, and hot flashes in midlife women." 94(3): 1037-1043.

Almeneessier, A. S., A. A. BaHammam, M. Alzoghaibi, A. H. Olaish, S. Z. Nashwan and A. S. BaHammam (2019). "The effects of diurnal intermittent fasting on pro-inflammatory cytokine levels while controlling for sleep/wake pattern, meal composition, and energy expenditure." PLOS ONE 14(12): e0226034.

Aswathappa, J., S. Garg, K. Kutty, and V. Shankar (2013). "Neck circumference as an anthropometric measure of obesity in diabetics." North American journal of medical sciences 5(1): 28-31.

BaHammam, A. S., and A. S. Almeneessier (2020). "Recent Evidence on the Impact of Ramadan Diurnal Intermittent Fasting, Mealtime, and Circadian Rhythm on Cardiometabolic Risk: A Review." Frontiers in Nutrition 7(28).

Bahijri, S., A. Borai, G. Ajabnoor, A. Abdul Khaliq, I. AlQassas, D. Al-Shehri, and G. Chrousos (2013). "Relative metabolic stability, but disrupted circadian cortisol secretion during the fasting month of Ramadan." PLoS One 8(4): e60917.

Beaton, E., J. Wright, G. Devenish, L. Do, and J. Scott (2018). "Relative validity of a 24-h recall in assessing intake of key nutrients in a cohort of Australian toddlers." Nutrients 10(1): 80.

De Rui, M., N. Veronese, F. Bolzetta, L. Berton, S. Carraro, G. Bano, C. Trevisan, S. Pizzato, A. Coin, E. Perissinotto, E. Manzato and G. Sergi (2017). "Validation of bioelectrical impedance analysis for estimating limb lean mass in free-living Caucasian elderly people." Clinical Nutrition 36(2): 577-584.

Faris, M., H. A. Jahrami, J. Alsibai, and A. A. Obaideen (2020). "Impact of Ramadan diurnal intermittent fasting on the metabolic syndrome components in healthy, non-athletic Muslim people aged over 15 years: a systematic review and meta-analysis." British Journal of Nutrition 123(1): 1-22.

Faris, M., M. I. Madkour, A. K. Obaideen, E. Z. Dalah, H. A. Hasan, H. M. Radwan, H. A. Jahrami, O. Hamdy, and M. G. Mohammad (2019). "Effect of Ramadan diurnal fasting on visceral adiposity and serum adipokines in overweight and obese individuals." Diabetes research and clinical practice.

Faris, M. e. A.-I., H. Jahrami, A. BaHammam, Z. Kalaji, M. Madkour, and M. Hassanein (2020). "A systematic review, meta-analysis, and meta-regression of the impact of diurnal intermittent fasting during Ramadan on glucometabolic markers in healthy subjects." Diabetes Research and Clinical Practice 165: 108226.

Foster, E., C. Lee, F. Imamura, S. E. Hollidge, K. L. Westgate, M. C. Venables, I. Poliakov, M. K. Rowland, T. Osadchiy, J. C. Bradley, E. L. Simpson, A. J. Adamson, P. Olivier, N. Wareham, N. G. Forouhi, and S. Brage (2019). "Validity and reliability of an online self-report 24-h dietary recall method (Intake24): a doubly labelled water study and repeated-measures analysis." Journal of nutritional science 8: e29-e29.

Karaagaoglu, N., and S. Yucecan (2000). "Some behavioral changes observed among fasting subjects, their nutritional habits, and energy expenditure in Ramadan." International journal of food sciences and nutrition 51(2): 125.

Kul, S., E. Savaş, Z. A. Öztürk, and G. Karadağ (2014). "Does Ramadan fasting alter body weight and blood lipids and fasting blood glucose in a healthy population? A meta-analysis." J Relig Health 53(3): 929-942.

Ling, C. H. Y., A. J. M. de Craen, P. E. Slagboom, D. A. Gunn, M. P. M. Stokkel, R. G. J. Westendorp, and A. B. Maier (2011). "Accuracy of direct segmental multi-frequency bioimpedance analysis in the assessment of total body and segmental body composition in middle-aged adult population." Clinical Nutrition 30(5): 610-615.

Madkour, M. I., A. T. El-Serafi, H. A. Jahrami, N. M. Sherif, R. E. Hassan, S. Awadallah, and M. Faris (2019). "Ramadan diurnal intermittent fasting modulates SOD2, TFAM, Nrf2, and sirtuins (SIRT1, SIRT3) gene expressions in subjects with overweight and obesity." diabetes research and clinical practice 155: 107801.

Muller, A. F., S. W. Lamberts, J. A. Janssen, L. J. Hofland, P. V. Koetsveld, M. Bidlingmaier, C. J. Strasburger, E. Ghigo, and A. J. Van der Lely (2002). "Ghrelin drives GH secretion during fasting in man." Eur J Endocrinol 146(2): 203-207.

Shadman, Z., N. Poorsoltan, M. Akhoundan, B. Larijani, M. Soleymanzadeh, C. Akhgar Zhand, Z. A. Seyed Rohani, and M. Khoshniat Nikoo (2014). "Ramadan Major Dietary Patterns." Iranian Red Crescent Medical Journal 16(9).

Sun, Y., D. L. Roth, C. S. Ritchie, K. L. Burgio, and J. L. Locher (2010). "Reliability and predictive validity of energy intake measures from the 24-hour dietary recalls of homebound older adults." Journal of the American Dietetic Association 110(5): 773-778.

Xing, Y., J. Liu, J. Xu, L. Yin, L. Wang, J. Li, Z. Yu, F. Li, R. Gao, and J. Jia (2015). "Association between Plasma Leptin and Estrogen in Female Patients of Amnestic Mild Cognitive Impairment." Disease Markers 2015: 1-5.

---

## [Editor Report · Decision Letter 1]

6 Aug 2020

Effect of diurnal intermittent fasting during Ramadan on ghrelin, leptin, melatonin, and cortisol levels among overweight and obese subjects: A prospective observational study

PONE-D-20-01785R1

Dear authors

We’re pleased to inform you that your manuscript has been judged scientifically suitable for publication and will be formally accepted for publication once it meets all outstanding technical requirements.

Kind regards,

Nayanatara Arun Kumar

Academic Editor

PLOS ONE

Additional Editor Comments (optional):

Dear authors

My sincere aplogize for the delay in the decision .The reviewers commentes has been done and rectified in the manuscript . So this manuscript can be accepted in this prestigious journal.Congragulations to you all

with regards and Best wishes

Dr. Nayanatara Arun Kumar

Associate Profeesor in Physiology

Kasturba Medical COllege, Mangalore